# STAT5A and STAT5B—Twins with Different Personalities in Hematopoiesis and Leukemia

**DOI:** 10.3390/cancers11111726

**Published:** 2019-11-04

**Authors:** Barbara Maurer, Sebastian Kollmann, Judith Pickem, Andrea Hoelbl-Kovacic, Veronika Sexl

**Affiliations:** Institute of Pharmacology and Toxicology, University of Veterinary Medicine, 1210 Vienna, Austria; sebastian.kollmann@vetmeduni.ac.at (S.K.); judith.pickem@gmail.com (J.P.); Andrea.hoelbl@vetmeduni.ac.at (A.H.-K.); veronika.sexl@vetmeduni.ac.at (V.S.)

**Keywords:** STAT5A, STAT5B, hematopoietic stem cells, STAT5B^N642H^, STAT5 mouse models, BCR–ABL, leukemia, hematopoiesis

## Abstract

The transcription factors STAT5A and STAT5B have essential roles in survival and proliferation of hematopoietic cells—which have been considered largely redundant. Mutations of upstream kinases, copy number gains, or activating mutations in *STAT5A,* or more frequently in *STAT5B,* cause altered hematopoiesis and cancer. Interfering with their activity by pharmacological intervention is an up-and-coming therapeutic avenue. Precision medicine requests detailed knowledge of STAT5A’s and STAT5B’s individual functions. Recent evidence highlights the privileged role for STAT5B over STAT5A in normal and malignant hematopoiesis. Here, we provide an overview on their individual functions within the hematopoietic system.

## 1. Introduction

The transcription factors signal transducer and activator of transcription 5A (STAT5A) and 5B (STAT5B) are part of the highly conserved Janus kinase (JAK)/STAT signaling pathway. They fulfill critical functions in processes like proliferation, differentiation, survival, and senescence. Triggers for JAK activation come from the stimulation of upstream membrane receptors which respond to cytokines or growth factors. Upon activation, JAK family members phosphorylate STAT5A/B on a critical tyrosine residue (pYSTAT5A/B), which induces a conformational change to parallel STAT5A/B dimers, exposing the DNA binding domain. After nuclear import, gene transcription is typically initiated at gamma interferon-activated sequence (GAS) motifs [1,2]. Beside the “classical” canonical signal transduction pathway, STAT5A/B can function via tyrosine phosphorylation-independent mechanisms (unphosphorylated STAT5A/B, uSTAT5A/B). As uSTATs, they seem to have a more global role by interacting with epigenetic and chromatin modifiers [3,4,5].

STAT5A/B signaling is enhanced in diverse hematopoietic cancers and is believed to drive disease. Enhanced STAT5A/B activation is achieved by copy number gains, enhanced protein expression, or gain-of-function (GOF) mutations, leading to higher pYSTAT5A/B levels contributing to tumor cell survival and disease progression. As such, STAT5A and STAT5B are in the focus of current pharmaceutical research [6]. Activating mutations occur much more frequently in *STAT5B* than in *STAT5A*, the underlying reason being widely enigmatic. The recent evidence has provoked studies that provide insights into specific roles of STAT5A and STAT5B in hematopoiesis, immune cell functions, and leukemogenesis—knowledge needed for future drug development approaches. This review focuses on the specific roles of STAT5A and STAT5B in different hematopoietic cell types and their impact on hematopoietic malignancies and treatment options.

## 2. Differences and Similarities of STAT5A and STAT5B

*STAT5A* and *STAT5B* arose approximately 310 to 130 million years ago in the course of early eutherian evolution. While birds and several other animals harbor one STAT5 gene, the co-presence of *STAT5A* and *STAT5B* became a mammal-specific feature [7,8], along with STAT5A’s master regulatory function in the mammary gland [9]. Interestingly, zebrafish developed an independent duplication of the *stat5* gene, consisting of *stat5.1* and *stat5.2* [10]. The intron–exon structure of the zebrafish and the murine isoforms are highly concordant. In contrast to the mammalian isoforms, zebrafish *stat5.1* and *stat5.2* are located on different chromosomes. *Stat5.1* is highly homologous to mammalian *STAT5A* and *STAT5B*, while *stat5.2* lacks a mammalian orthologue [11,12]. Mutated *stat5.1* zebrafish displayed a reduced body size, in line with reduced growth hormone (*gh)1* mRNA levels and Stat5.1 binding to the *gh1* promoter, while *stat5.2*-mutated zebrafish showed no developmental defect [13].

Initially, STAT5A/B was described as a prolactin-responsive DNA binding protein in mammary epithelial cells [14,15]. Soon afterwards, it was found that interleukin (IL)-2, IL-3, and erythropoietin (EPO) signaling activate the protein by tyrosine phosphorylation and that it exists in two flavors—STAT5A and STAT5B [16,17,18,19]. *STAT5A* and *STAT5B* are encoded by two juxtaposed genes with the transcriptional start sites within 10.7 kb of each other, mapping to chromosome #17 in humans and to chromosome #11 in mice. They are translated to two more than 90% homologous proteins differing primarily at their C-termini [20] (see Appendix A). Similar to other STAT proteins, STAT5A and STAT5B consist of six functional domains (Figure 1): The N-terminus is important for oligomerization, and the C-terminus contains the phosphorylation sites involved in STAT5A/B activation [21,22,23,24]. Comparing their protein structures, STAT5A has 12 amino acids more on the C-terminus. The last 20 amino acids of STAT5A and the last 8 amino acids of STAT5B are unique to the respective proteins. STAT5A differs in one residue and lacks 5 residues between the Src-homology 2 (SH2) and transactivation domain, the so-called phosphotyrosyl tail [25,26], depicted in Figure 1 and Appendix A. These differences may account for the non-redundant roles of STAT5A and STAT5B by affecting gene regulation or specific protein–protein interactions [27,28]. The DNA binding domain differs by five amino acids which contribute to homodimer-specific DNA binding affinities [25]. These individual DNA binding specificities of pYSTAT5A/B homo- or heterodimers may influence the transcription of target genes [25,29], but the formation of pYSTAT5A/B homo- and heterodimers was suggested to occur randomly [30]. Different STAT5A/B expression levels, cytokine receptor affinities, and oligomerization properties are further factors probably influencing the signaling response in each cell type.

STAT5A/B functions are modified via post-translational modifications at different sites (Figure 1). The critical tyrosine phosphorylation sites for activation are Y694 in STAT5A and Y699 in STAT5B [31]. In addition, serine phosphorylation at S726 and S780 for STAT5A (corresponding mouse serine phosphorylation sites S725 and S779) and at S715 and S731 for STAT5B enables enhanced activation and nuclear translocation [32,33]. STAT5A contains two additional phosphorylation sites: STAT5A S127/S128 involved in ERB4-mediated activation; and STAT5A T682/T683 associated with IL-3 signaling [34,35]. STAT5B comprises additional phosphorylation sites taking part in inducing or inhibiting transcription, e.g., S193 is associated with mTOR kinase activity [36,37,38,39,40]. Known upstream kinases for serine phosphorylation are the MAPK family, ERKs, JNK, p38 MAPK, PAK kinases in a RHO/RAC dependent manner, and CDK8. The latter was associated with enhanced mediator complex occupancy at its target genes [32,41,42]. Additionally, STAT5B tyrosine phosphorylation sites Y725, Y740, and Y743 were described to be highly induced by epidermal growth factor (EGF) stimulation. While Y740 and Y743 were reported as negative regulators of transcription by reducing Y699 phosphorylation, Y725 displayed a much weaker effect with controversial transcriptional contributions [40,43,44]. STAT5B also contains SUMOylation (inhibiting STAT5 phosphorylation) and acetylation (promotes STAT5 phosphorylation) sites—lysine acetylation may even be a prerequisite for efficient STAT5 dimerization, translocation, and activation of transcription [45,46,47]. O-GlcNAcylation of STAT5A’s T92 was described to enhance tyrosine phosphorylation and, consequently, transactivation [48].

A different mode of action of STAT5A/B is added by non-canonical functions of uSTAT5 first shown in *Drosophila* [49]. In a colon cancer model, uSTAT5A stabilized heterochromatin by binding to heterochromatin protein 1α (HP1α) and suppressing the “cancer expression signature” [3]. In hematopoietic progenitor cells, uSTAT5 prevented megakaryocyte differentiation [5], as discussed below. A very recent study focusing on uSTAT5A and uSTAT5B in acute myeloid leukemia (AML) suggested that uSTAT5B is a key regulator of differentiation of AML cells. Isoform-specific interaction partners were identified in AML cell lines: uSTAT5A interacts with DBC1, while uSTAT5B interacts with ETV6 [50].

Various activating and repressing interactions with transcriptional co-factors and epigenetic modulators have been described for STAT5A/B, which have been recently reviewed [51]. In the following, we focus on our current understanding of STAT5A and STAT5B functions in the differentiation of hematopoietic lineages.

## 3. STAT5A/B Deficiency in Mice and Men

To understand the roles of STAT5A/B, genetically engineered mice were generated (Table 1). First insights were derived by *Stat5a/b^ΔN^* mice, which expressed truncated N-termini of STAT5A and STAT5B [52,53,54,55,56]. STAT5A/B^ΔN^ proteins still formed dimers and bound DNA, but tetramer formation and complete target gene transcription were significantly impaired [21]. Hematopoiesis in *Stat5a^ΔN^*, *Stat5b^ΔN^,* and *Stat5a/b^ΔN^* mice was affected to a minor degree [56]. Likewise, tetramer formation was blocked in the *Stat5a/b^DKI^* (double knock-in) mouse model, in which mutations were introduced into the N-termini of *Stat5a* or *Stat5b.* Both mouse models showed reduced numbers of natural killer (NK) cells, while T cell numbers were exclusively reduced in *Stat5a/b^ΔN^* mice [57,58].

The complete genetic abrogation of STAT5A and STAT5B (*Stat5a/b*^−/−^*)* resulted in perinatal lethality; the few survivors displayed severe microcytic anemia, reduced numbers of CD8^+^ T cells, and a block in the pre–pro-B cell stage. The anemia was explained by apoptosis of fetal liver cells and reduced expression of iron-regulatory protein 2 (IRP2) and transferrin receptor 1 (TFR1) [52,59,64]. To study tissue-specific STAT5A/B functions or allow conditional deletion, *Stat5a/b* floxed mice [59] were crossed with *Mx1*-Cre, *vav1*-Cre, or *Tie2*-Cre mice to elucidate STAT5A/B’s function in the hematopoietic system [60,61,62]. As the prenatal lethality of *Stat5a/b*^−/−^ mice was connected to severe combined immunodeficiency, erythroid defects, and subsequent anemia [52,64,65], it was somehow surprising that hematopoietic-specific STAT5A/B deletion led to anemia and lymphopenia, but did not influence survival [60,61].

Moreover, mouse models either lacking STAT5A [9] or STAT5B [63] gave insight into gene-specific functions. *Stat5a*^−/−^ females showed defective mammary gland formation and failed to lactate, while *Stat5b*^−/−^ mice were smaller, showed altered GH signaling, and displayed stronger hematopoietic defects [9,63,66]. “Compensatory” mechanisms due to the absence of STAT5A or STAT5B in the whole organism cannot be ruled out. Murine models with floxed loci of either *Stat5a* or *Stat5b* are so far unavailable.

Humans with STAT5B deficiency suffer from a rare autosomal disorder resulting in dwarfism, prominent forehead, eczema, and a high-pitched voice. In line with the role of STAT5B as mediator of IL-2 signaling, these patients undergo recurrent infections due to immunodeficiency caused by impairment in T, regulatory T (T_reg_), and NK cell differentiation and activation. The homozygous mis- or nonsense mutations in these patients lead to non-detectable STAT5B expression [67,68,69,70,71].

Recently, dominant negative germline mutations of *STAT5B* were discovered in patients. Here, the wild-type (wt)–mutant heterodimers fail to translocate to the nucleus or bind DNA. This leads to growth failure and hyper-IgE syndrome [72].

Importantly, STAT5B deficiency-associated diseases confirm that STAT5A does not compensate for all functions of STAT5B. Until now, STAT5A deficiency in humans has not been reported. One may speculate that this provokes only a very mild or absent phenotype, or—the opposite—its absence would be fatal.

## 4. STAT5B as the Dominant Player in Hematopoietic Lineages

STAT5A/B are fundamental for myelopoiesis, lymphoid development, macrophage functions, megakaryopoiesis, basophil, eosinophil, and mast cell functions [73,74,75]. This is explained by STAT5A/B’s function as key signaling molecules downstream of various cytokine and growth factor receptors, e.g., IL-2, -3, -4, -5, -7, -9, -13, -15, -21, EPO, thrombopoietin (TPO), GH, prolactin, stem cell factor (SCF), Flt3, granulocyte-macrophage (GM) colony-stimulating factor (CSF) or GCSF [73,76]. In all differentiated hematopoietic cell types, STAT5B is expressed at higher levels compared to STAT5A (Figure 2) [77].

### 4.1. STAT5A/B as Regulators of Erythropoiesis

The importance of STAT5A/B in erythroid differentiation downstream of EPO/EPOR/JAK2 signaling has been well established [78,79]. Hematopoietic deletion of STAT5A/B resulted in anemia, defining STAT5A/B as regulator of iron uptake (control of TFR1 expression) and survival genes in erythroid cells [60,61]. Their essential role was demonstrated by *Stat5a^S710F^* (cS5^F^, a hyperactive *Stat5a* variant) expression in *Epor*^−/−^ or *Jak2*^−/−^ fetal liver cells, enabling self-renewal and erythroid differentiation [80]. Overexpression of cS5^F^ in human CD34^+^ cells induced erythroid differentiation [81]. In an elegant experimental set-up, Villarino and colleagues determined the specific functions of STAT5A and STAT5B in single-allele expressing mice and found no difference in the hematocrit of these mice [66], suggesting a redundant role of both genes in erythroid development. Transgenic expression of cS5^F^, STAT5B, or hyperactive STAT5B^N642H^ under the hematopoietic *vav* promoter [82] did not affect hematocrit levels [83,84], which points to a strictly controlled regulation of pYSTAT5A/B signaling in erythrocytes.

### 4.2. Megakaryopoiesis—Non-Canonical STAT5A/B Prevent Differentiation

TPO activates the megakaryocytic differentiation program via JAK2-dependent pYSTAT5A/B activation, regulating, e.g., *Bcl-xl* expression and cell survival [85]. Human CD34^+^ cells differentiated to megakaryocytes upon STAT5A/B downregulation in line with the prevention of megakaryocyte development by activated STAT5A expression (cS5^F^) [81].

A non-canonical function for nucleus-located uSTAT5 was described in megakaryocyte differentiation: uSTAT5 bound to CTCF binding sites and suppressed differentiation by antagonizing ERG in the absence of TPO. Upon TPO stimulation, STAT5 was tyrosine phosphorylated and redistributed to canonical GAS sites [5]. Based on the knockdown experiments of Park et al., this non-canonical role was mainly assigned to STAT5B—at least in the studied experimental system (HPC-7 cells). Further evidence stems from enforced expression of STAT5B^Y699F^, a mutant incapable of getting phosphorylated: Upon stimulation with TPO, the mutant protected uSTAT5-bound enhancers from deacetylation. This observation underlined the role of uSTAT5 in maintaining regulatory elements [4]. Of note, uSTAT5A was found to participate in chromatin compaction by binding to HP1α [3]—so far not reported in hematopoietic cells. Further studies are needed to determine whether similar functions of nuclear u- and pY-STAT5A or -STAT5B, respectively, are important for the chromatin landscape and consequently lineage determination.

### 4.3. STAT5A/B Promote Survival and Differentiation of B Cells

The IL-2 family cytokines, characterized by signaling through the common gamma chain, regulate STAT5A/B activation and play important parts in the immune system [2]. In the few surviving *Stat5a/b*^−/−^ mice, B cell development was blocked in the pre–pro-B cell stage [52]. This block was explained by the absence of *Mcl-1*, a direct STAT5 target gene promoting survival during early B cell stages [86]. IL-7/STAT5A/B signaling is essential for B cell development—IL-7R signaling-deficient mice were blocked at the earliest stages of B cell development and lacked mature B cells in the periphery. A constitutive active (ca) *Stat5b* rescued B cell development by upregulating pro-survival genes [54,86,87], and ca STAT5B (*Stat5b*-CA-tg) transgenic mice had an increased number of pro-B cells [88]. In *Stat5a^+/^*^−^*; Stat5b*^−/−^ mice, B cell numbers were increased and auto-antibodies were enriched, a phenotype more modest in *Stat5a*^−/−^*; Stat5b^+^*^/^^−^ mice [66].

In *Stat5b*-CA-tg mice, STAT5B bound to genes involved in normal B cell development like pre-B cell receptor (BCR) genes (*Syk*, *Blk*, *Blnk*, *Carma1*, *Irf4*, *Irf8*, or *Ikaros*) and blocked B cell differentiation. This was considered to contribute to transformation in B cell acute lymphocytic leukemia (B-ALL) [89]. Whether this is an exclusive function of STAT5B or whether it is shared by STAT5A remains to be determined.

### 4.4. STAT5B is the Major Player in NK Cells

NK cells represent an important part of the innate immune system and function as immediate effector cells against viral infections, pathogens, and malignant cells. They also depend on IL-2 family cytokines—especially IL-2 and IL-15 signaling are essential in NK cells. STAT5A/B is a master regulator of NK cell proliferation, survival, and cytotoxicity [2,90,91,92]. Accordingly, NK cells were grossly absent in *Stat5a/b*^−/−^ [65], as well as NK cell-specific *Stat5a/b*^−/−^ [93] and *Jak1*^−/−^ mice [94]. NK cell survival was rescued by overexpression of BCL-2 [95]. STAT5A/B dimers were sufficient for NK cell development, whereas tetramers were needed for maturation [96].

Murine and human data pinpoint to a key role of STAT5B in NK cells: STAT5B deficiency in mice resulted in a more pronounced reduction of NK cell numbers, activity after IL-2 and IL-15 stimulation, and cytolytic function compared to STAT5A [97,98]. This may be explained by the higher expression levels of STAT5B compared to STAT5A in NK cells (Figure 2).

Loss-of-function (LOF) mutations in *STAT5B* led to human primary immune deficiencies affecting NK cells [99]—so far not reported for *STAT5A* [91]. Interestingly, a chemical-induced mutation in the linker domain of murine *Stat5a* led to reduced STAT5A levels and negatively influenced NK cell development, maturation and activation [100]. These results argue for the need of correct stochiometric ratios of STAT5A and STAT5B to generate complete NK cell functionality,

### 4.5. CD8^+^ T Cells are Sensitive to Elevated pYSTAT5A/B Levels

STAT5A and STAT5B proteins are essential downstream mediators of IL-2R and IL-7R signaling to regulate T cell differentiation [101]. The reduced number of thymocytes in *Stat5a/b*^−/−^ mice resulted in a severely decreased number of CD8^+^ T cells, loss of γδ T cells [52] and a higher proportion of CD4^−^CD8^−^ thymocytes [65]. Deletion of STAT5A/B at the CD4^+^CD8^+^ T cell differentiation stage also resulted in a massive reduction of CD8^+^ T cells [52]. Differentiation of CD8^+^ T cells is regulated by STAT5A/B in a dose-dependent manner [102,103] and STAT5A/B-tetramers are required for expansion of antigen-specific activated CD8^+^ T cells [57]. Interestingly, IL-7R-mediated STAT5A/B signaling upregulated *Runx3*, *Bcl-2,* as well as *Mcl1* mRNA expression, which allowed to bypass T cell receptor signaling to induce CD8^+^ T cell differentiation [104].

*Stat5b* deficiency led to a more pronounced CD8^+^ T cell reduction compared to *Stat5a* loss [66]. Vice versa, overexpression of STAT5B led to an increase in CD8^+^ and γδ T cells [84,105]. Expression of cS5^F^ in CD8^+^ T cells enhanced effector and memory CD8^+^ T cell survival [106,107] and its broad hematopoietic expression induced CD8^+^ T cell leukemia [83].

### 4.6. CD4^+^ T Cell Development—Quantities Matter

CD4^+^ T cells can be further subdivided into T helper (T_h_) 1 cells, T_h_2 cells, T_reg_s, follicular helper T cells, T_h_9, T_h_17, and T helper type GM-CSF cells whose differentiation is induced by specific cytokines. For all of these cell types, STAT5A/B signaling contributes to differentiation, function, or survival, which has been recently reviewed [108].

STAT5B deficiency in mice had a greater impact on CD4^+^ T cell numbers compared to STAT5A deficiency. This difference was rather explained by higher expression levels of STAT5B in CD4^+^ T cells than by differences in DNA binding site occupancy [66]. These data generated from knockout mouse models do not completely reflect the insights gained from human cells: Knockdown of STAT5B in primary human CD4^+^ T cells reduced expression of IL-2Rα and FOXP3—the main markers for T_reg_s—to a greater extent than deletion of STAT5A. In contrast, BCL-X expression was affected primarily by STAT5A knockdown [109]. In a global ChIP-Seq approach in human CD4^+^ T cells, Kanai et al. confirmed the preferential occupation of *IL2RA* and *FOXP3* by STAT5B after IL-2 stimulation, and further showed STAT5B binding to *DOCK8* and *SNX9*, both functioning in T cell immune responses. Exclusive STAT5A binding sites were described to take part in neural development and function. Common STAT5A and STAT5B binding sites were found at proliferation and survival genes implicating redundant functions in these processes [27].

Despite many redundant roles of STAT5A and STAT5B in T and NK cells, the higher expression levels of STAT5B define it as the prominent isoform in immune cells.

## 5. STAT5A/B are Required for Hematopoietic Stem Cell Maintenance and Self-Renewal

Hematopoietic stem cells (HSCs) are defined by their ability to reconstitute the hematopoietic tree with blood cells of all lineages, while maintaining the ability to produce a multipotent HSC by self-renewal. To test the role of STAT5A/B in the repopulation capacity of stem cells, competitive and non-competitive transplantations of wt, *Stat5a/b^∆N^* and STAT5A/B-deficient bone marrow (BM) or fetal liver cells were performed. *Stat5a/b^∆N^* cells exert a drastic reduction in the ability to reconstitute the hematopoietic system [110,111,112]. Upon conditional or hematopoietic-specific *Stat5a/b* deletion, this defect was recapitulated by a depletion of the long term (LT)-HSC pool [60,62]. In line with these results, RNAi-mediated downregulation of STAT5A/B resulted in decreased long-term expansion capacity of human progenitor cells [113,114]. Collectively, these data demonstrate a role for STAT5A/B in HSC maintenance and self-renewal.

Bunting and colleagues linked the defect in LT-HSC maintenance of *Stat5a/b*^−/−^ cells to increased apoptosis and loss of quiescence. Quiescence genes like *Tie2*, *Mpl*, *Slamf1*, or *Cited2* were downregulated in HSCs derived from STAT5A/B-deficient BM transplants. They also showed that the *Slamf1* locus is directly bound by STAT5A/B. In addition, the TPO-induced HSC-related genes *Tie-2* and *p57* were downregulated in cells lacking STAT5A/B [60,62]. A recent study employed single-cell qPCR to study the deregulation of several quiescence- and HSC-associated genes in STAT5A/B-deficient LSKs and LT-HSCs. Downregulation of quiescence genes like *Mpl*, *Tie2,* or *Cited2* in HSCs derived from STAT5A/B-deficient mice was confirmed. Loss of STAT5A/B in LT-HSCs induced myeloid and lymphoid–myeloid multi-lineage priming based on mRNA expression profiles [115], assigning STAT5A/B as a keeper of HSC quiescence.

Using an inducible system in CD34^+^ human cord blood cells, the induction of STAT5A/B activity provided an advantage in long-term proliferation of HSCs, but not in multi-lineage progenitors [116]. This effect was even enhanced upon down-modulation of GATA1 and allowed the identification of GATA1 (erythroid committed)-independent STAT5A/B target genes [117]. Besides well-known target genes like *Pim1* or *Osm*, STAT5A/B directly bound the HIF2α promoter and induced gene transcription. As HIF2α is critical for glucose uptake, this observation suggests a major role for STAT5A/B in maintaining self-renewal under hypoxic conditions [116].

Moreover, STAT5A/B regulates the expression of miR-193b, which controls expansion of HSCs by reducing c-KIT expression. This c-KIT reduction inhibits cytokine-induced STAT5A/B and AKT signaling and prevents uncontrolled HSC expansion [118]. Of interest, mice lacking the typical STAT5 target gene *Pim1* failed to reconstitute lethally irradiated recipient mice. *Pim1* activity regulates CXCR4 expression, suggesting an important role for STAT5A/B in homing and migration of HSCs [119].

Despite the profound role of STAT5A/B in HSC renewal, quiescence, and lineage differentiation, none of these studies focused on the individual roles of STAT5A and STAT5B in HSC biology. In contrast to differentiated hematopoietic cells, mRNA levels of *STAT5A* and *STAT5B* are comparable in HSCs (Figure 2).

As mentioned, numbers of LT-HSCs were reduced in STAT5A/B-deficient mouse models [62], while they were drastically increased in transgenic STAT5B^N642H^ and STAT5B wild-type mice [84]. Further evidence stems from the expression of recombinant oncogenic STAT5A or STAT5B variants in HSCs and progenitors. Here, STAT5A S779 phosphorylation fine-tuned proliferation and transformation of HSCs and progenitors [120]. Given the comparable expression levels of STAT5A and STAT5B in HSCs, it remains to be determined whether and how they induce the same set of target genes, which is currently enigmatic.

A further layer of complexity is provided by the interferon (IFN) signaling pathway that mediates cell cycle induction in LT-HSCs [121,122]. IFNs first activate dormant LT-HSCs, and later on cause them to re-enter their quiescent state to avoid apoptosis and DNA damage [123]. LT-HSCs which have once experienced IFNs show a reduced potential of reconstitution even if they have regained their quiescent state. These observations are in line with those of STAT5A/B deficiency [124].

One potential mechanism, how STAT5A/B interferes with the decreased repopulation capacity upon IFN signaling, might be via SOCS1 upregulation. SOCS1 negatively regulates the levels of pYSTAT5A/B. It impairs TPO signaling, which finally ends up in lower HSC self-renewal and reconstitution [125,126,127]. Vice versa, activated STAT5B may repress IFN-α/β and IFN-γ signaling in HSCs, as has been recently demonstrated in transformed pro-B cells [128].

## 6. STAT5A/B as Oncogenes in Hematopoietic Cancer

STAT5A/B are deregulated in a variety of hematopoietic and non-hematopoietic tumors. Amongst others, ALL, myeloproliferative neoplasms, AML, chronic myeloid leukemia (CML), B-ALL, and peripheral T cell leukemia/lymphoma (PTCL) show enhanced STAT5A/B signaling [129,130]. STAT5A and STAT5B act as proto-oncogenes by regulating proliferation and survival [131,132]. They directly promote transcription of anti-apoptotic genes like *Mcl-1, Bcl-2, Bcl-x_L_*, *miR15/16* or *C-Myc*, *D-type cyclins D1*, *D2* and *D3*, cytokines/cytokine receptor chain expression exemplified by OSM, IL-2Rα, IL-4Rα or IL-7Rα, and are associated with growth factor receptor signaling, contributing to many essential functions in cancer [110,132,133,134,135,136]. STAT5A/B activation is, in most cases, induced by hyperactive upstream tyrosine kinases (TK) (e.g., JAK2^V617F^, BCR-ABL, FLT3-ITD, KIT^D816V^). Recurrent somatic point mutations in *STAT5B* in mature NK/T cell neoplasms recently concentrated research on STAT5B mutations and how they drive disease [84,137].

### 6.1. STAT5A and STAT5B Mutations as Disease Drivers

Although redundant functions have been assigned to STAT5A and STAT5B in T cells [66,84], the *STAT5B* gene is predominantly affected in NK/T cell neoplasia by point mutations localized mainly in the SH2 domain. These GOF mutations lead to enhanced parallel dimerization, nuclear translocation, gene regulation, and persistence against dephosphorylation. Examples for *STAT5B* mutations are N642H, G596V, Y665F, T648S, or T628S [138], of which some have been analyzed in vitro [84,137,139,140,141,142].

The most recurrent mutation, *STAT5B*^N642H^, was detected across many forms of PTCL [139,140,141,143,144,145,146,147,148] and has also been reported in myeloid neoplasia with eosinophilia [149], as well as neutrophilic leukemia [150].

The STAT5B^N642H^ mutation stabilizes dimer formation and leads to increased phosphotyrosine levels [139,140,141,143,144,145,146,147,148]. Confirmation stems from the recently published crystal structure of *STAT5B*^N642H^: Hyperactivation is explained by an “open” SH2 domain state, which allows facilitated access to the peptide binding pocket [137]. However, upstream cytokine signaling is still a prerequisite for the activation of STAT5B [84,143,151,152].

Transgenic mouse models expressing either wt *Stat5b,* or ca *Stat5a* or ca *Stat5b* in hematopoietic lineages (summarized in Table 2) were used to study the role of STAT5A/B in leukemia. When expressing high levels of cS5^F^ under the hematopoietic *vav* promoter, mice developed CD8^+^ T cell leukemia/lymphoma [83]. The first transgenic mouse model expressing *STAT5B^N642H^* in the hematopoietic system (*vav* promoter) developed an aggressive CD8^+^ T cell leukemia with organ infiltrations by CD8^+^, CD4^+^, and γδ T cells [84,137]. Transplantation models derived from this transgenic mouse verified the oncogenic role for STAT5B^N642H^ in NKT [152] and γδ T cells [137].

Despite the phenotypic similarities of the cS5^F^
*vav* mouse model compared to the *STAT5B^N642H^* transgenic mice, the latter disease model is far more drastic and aggressive. This pinpoints to a greater oncogenic potential of *STAT5B* compared to *STAT5A*. The higher number of deregulated genes in STAT5B^N642H^- compared to STAT5A^S710F^-mutated CD8^+^ T cells supports this concept. Interestingly, both mutations resulted in exclusive sets of deregulated genes pinpointing to specific functions [83]. In addition, certain γδ T cell subsets reacted differently—STAT5B^N642H^ supported IFN-γ-producing CD27^+^ γδ T cells, whereas expression of STAT5A^S710F^ led to expansion of IL-17-producing γδ T cells—explained by an inverse regulation of *Tbet* [156]. In summary, both transgenic mouse models verified the privileged role of STAT5A/B signaling in CD8^+^ T cells and the high sensitivity to altered pYSTAT5A/B levels [83,84].

So far, novel DNA binding sites or interaction partners of STAT5B^N642H^ have not been thoroughly analyzed. In a T-ALL model, co-operative HOXA9/STAT5B^N642H^ transcription enhanced the STAT5 transcriptional signature [157]. Decreased methylation of potential polycomb repressor complex 2 (PCR2) binding sites in STAT5B^N642H^-expressing CD8^+^ T cells and consequent upregulation of aurora kinases, known PRC2 target genes, suggest combinatorial treatments [84]. Understanding the consequences of STAT5B^N642H^ is a prerequisite to establish targeted treatments.

### 6.2. STAT5B—The Major Player Downstream of BCR–ABL

BCR–ABL^+^ leukemia is one of the best studied experimental model systems of a STAT5-dependent disease. BCR–ABL is a fusion protein with a potent and constitutive kinase activity. Almost 100% of all CML and ~30% of B-ALL cases are associated with the *t(9;22)(q34;q11)* reciprocal translocation resulting in the *Philadelphia chromosome*. Imatinib, a TK inhibitor targeting BCR–ABL, and its follow-up inhibitors improved the prognosis of CML patients incredibly, but treatment-resistant leukemic stem cells (LSCs) remain [158].

STAT5A/B acts as critical node in the signaling network downstream of BCR–ABL [52] and is indispensable for initiation and maintenance of BCR–ABL^+^ leukemia [159,160]. BCR–ABL is capable of directly or indirectly phosphorylating and activating STAT5A/B [161]. Even LSCs require STAT5A/B signaling—lowering STAT5A/B levels in an already established leukemia blocks the disease and disables BCR–ABL^+^ LSCs. Elevated levels of STAT5A/B contribute to a higher resistance rate to TK inhibitors in BCR–ABL^+^ leukemia [162]. The STAT5 target gene PIM2 contributes to imatinib resistance and its inhibition sensitized LSCs towards pharmacological treatment [163]. Suppression of the STAT5 target genes *PIM1* and *BCL2* (PIM kinase inhibitor AZD1208 and BCL2 antagonist Sabutoclax) induced apoptosis in BCR–ABL^+^ ALL cells [164].

Until recently, we lacked an understanding of whether and how STAT5A and STAT5B contribute individually to BCR–ABL-driven diseases. *STAT5A*-specific knockdown in human cells revealed no effect on survival, while *STAT5B*-diminished cells displayed increased levels of apoptosis and lost their self-renewal potential [35,165,166]. BCR–ABL directly activates STAT5B to a higher extent than STAT5A, as STAT5A remains partially in the cytoplasm [35]. Imatinib-resistant cell lines upregulate STAT5A; vice versa, cells are increasingly sensitive towards TK inhibitor-treatment upon knockdown of *STAT5A* [165].

Using BM of *Stat5a^−^*^/*−*^ and *Stat5b^−^*^/*−*^ mice, STAT5B was identified as the dominant isoform downstream of BCR–ABL, as it facilitates transformation via suppressing IFN-α/β and IFN-γ signaling. The relevance of this finding is supported by data from human patients suffering from *STAT5B*-GOF mutant PTCL. There, the picture was inverse: IFN signaling was downregulated upon STAT5B hyperactivation [128].

Disrupting the STAT5(B)–BCR–ABL interaction in STAT5-dependent hematopoietic diseases is of therapeutic relevance. Whether this defined role of STAT5B as predominant onco-protein extends also to other TK-driven malignancies (such as *JAK2^V617F^* or *FLT3-ITD*) remains to be elucidated.

### 6.3. STAT5A/B as Potential Opponents in NPM–ALK+ Lymphoma

The oncogenic fusion protein of anaplastic lymphoma kinase (ALK) with nucleophosmin 1 (NPM1) in anaplastic large cell lymphoma leads to the activation of multiple intracellular signal transduction pathways including PI3K–AKT, MAPK/ERK, mTOR, STAT3, and STAT5B [167,168]. NPM–ALK^+^ cells predominately express STAT5B, which controls proliferation and survival. Downregulation of STAT5A was explained by epigenetic silencing via methylation of its promoter by the NPM–ALK/STAT3 signaling axis. Forced expression of STAT5A led to downregulation of NPM–ALK through direct transcriptional inhibition [169]. This might indicate opposing roles, namely tumor-suppressive STAT5A and oncogenic STAT5B, in NPM–ALK-driven ALCL. Currently, NPM–ALK is the only oncogene for which an antagonistic function of STAT5A and STAT5B has been described.

## 7. Direct STAT5A/B Inhibition Remains Challenging

STAT5A/B hyperactivation is a common feature of hematopoietic malignancies, with point mutations being primarily reported for *STAT5B*. Due to its disease-driving role in various forms of myeloid and lymphoid leukemia/lymphomas, it represents a potential therapeutic target [170]. The lack of an enzymatic activity in the transcription factors STAT5A and STAT5B makes the development of specific inhibitors difficult. The structural similarity of STAT5A and STAT5B to each other, as well as to other STAT proteins, adds a further level of complexity. Here, we focus on direct STAT5 inhibitors, since inhibitors of upstream JAKs have been rigorously reviewed [171,172].

Although promiscuous with respect to their peptide binding motifs, a promising option is to target the SH2 domain of STAT5A/B to prevent tyrosine phosphorylation, activation, and nuclear translocation. The lead compound AC-3-19, whose structure is based on salicylic acid, turned out to not be potent enough for clinical translation [83,173,174]. Its further optimization led to AC-4-130, which was used successfully for in vitro and in vivo treatment of FLT3–ITD^+^ AML cell lines, as well as on primary AML cells. Its cytotoxic potential was enhanced by combinatorial treatment with the JAK1/2 inhibitor ruxolitinib or the p300/pCAF inhibitor garcinol [175]. The sterical confirmation of *STAT5B^N642H^* hinders AC-4-130 binding [175], making it not applicable for targeted treatment of patients who carry *STAT5B^N642H^*. Another example for a STAT5A/B SH2 domain inhibitor, IST5-002, blocked phosphorylation and nuclear translocation of STAT5A/B in BCR–ABL^+^ in vitro and in vivo systems [164,176]. Via a virtual compound library screening approach and further structural adaptions, the first STAT5A SH2 domain inhibitor was identified, showing a modest reduction of pYSTAT5A levels in BCR–ABL^+^ cells [177]. All compounds require further modifications to improve the bioavailability, stability, and potency.

A high selectivity for STAT5B-tyrosine phosphorylation inhibition over STAT5A was attributed to the catechol bisphosphate derivatives Capstafin [178] and Stafib-1 [179], which has been further modified to Stafib-2 [180]. STAT5B selectivity was assigned to a STAT5B-specific amino acid in the linker domain, which might represent a novel design approach [181]. So far, in vivo data are not available.

Nucleic acid-based approaches aiming to interfere with STAT5A/B DNA binding (e.g., dominant negative constructs, G-quartet oligonucleotides, decoy oligonucleotides, metal-based inhibitors) or STAT5A/B expression (antisense or siRNA) have been successful in vitro and in vivo [19,182,183,184]. What remains problematic in the clinic is delivery of these constructs to their preferred site [185]. Furthermore, cell-permeable peptides or mimetics have been identified, which bind to the protein or even the STAT5A/B DNA binding domain itself [186]. Permeability and stability of the peptides still represent a hurdle for clinical usage [187]. Recently, ATP was found to bind to and inhibit the STAT5B SH2 domain [188], which needs to be validated in cellular systems. Until now, none of these defined inhibitors met the requirements for entrance into clinical assessment.

Insights into mutation-specific changes in the transcriptome or methylome of *STAT5B*-GOF mutated cancer cells may open novel therapeutic avenues. Combinatorial treatments will be helpful to prevent resistance development and reduce side effects. In the transgenic STAT5B^N642H^ mouse model, RNA-seq analysis revealed that STAT5B^N642H^ CD8^+^ T cells upregulated cell cycle-driving and EZH2 target genes like *Top2A* and *Aurkb* [84]. Importantly, STAT5A/B and EZH2 have been shown to interact [189]. In STAT5B^N642H^ CD8^+^ T cells, DNA methylation was reduced at EZH2 and SUZ12 binding sites, both components of the chromatin remodeling complex PRC2 [190]. These observations indicate a competition of EZH2 and STAT5B^N642H^ for binding sites, leading to an altered transcriptome, including aurora kinases. Aurora kinases are important for cell division and represent promising targets in leukemia treatment. Specific inhibitors are currently under investigation in clinical trials [191,192,193]. Combining JAK and aurora kinase inhibitors resulted in the selective cell death of STAT5B^N642H^-expressing CD8^+^ T cells and NKTCL cell lines, which offers a targeted treatment option in STAT5B^N642H+^ PTCL patients [84,146]. Whether JAK/aurora kinase inhibition can effectively eradicate or ameliorate the STAT5B^N642H^-driven T cell disease in vivo remains to be shown.

Selective and effective low dosage STAT5A, STAT5B, or STAT5A/B inhibitors are not clinically available. Suppression of the immune system may be the downside of STAT5A/B inhibition comparable to the side effects of JAK inhibitors [194,195,196]. Similarly, anemia, thrombocytopenia, diarrhea, or neurotoxicity have to be explored in pre-clinical trials before new drug treatment options can be concluded. As the mode of action of STAT proteins is distinct from JAK action in many circumstances, side effects cannot be anticipated. Finally, combination therapies using epigenetic inhibitors targeting STAT5 cofactors, such as bromodomain and extra terminal inhibitors (BETi) and histone deacetylase inhibitors (HDACi), might offer alternative therapeutic approaches [197,198,199,200,201,202].

## 8. Conclusions

Until now, many studies focused on the collective roles of STAT5A and STAT5B in healthy and malignant hematopoiesis. Establishment of tools distinguishing between STAT5A and STAT5B will help to clarify redundant and non-redundant contributions in hematopoiesis and leukemogenesis. Only recently is a privileged role for STAT5B unveiling. The discoveries of *STAT5B* mutations in NK and T cell leukemia/lymphoma and *STAT5B-*deficient patients unambiguously indicate STAT5B’s particular importance. Its distinct role may stem from specific or preferential DNA binding sites and target gene expression, distinct protein–protein interactions, or non-canonical signaling (Figure 3). Exclusive targeting of STAT5B—while sparing STAT5A—is a great challenge. The most promising avenue to date encompasses the combined blockage of JAK with specific downstream targets of mutated STAT5B—as exemplified with aurora kinase inhibition. These downstream targets may be specific for the upstream TK mutation, the *STAT5B*-GOF mutation, and the affected cell type, which complicates clinical approaches. A more detailed understanding of STAT5A’s and STAT5B’s physiological roles will facilitate future clinical interventions in hematopoietic malignancies.

## Figures and Tables

**Figure 1 cancers-11-01726-f001:**
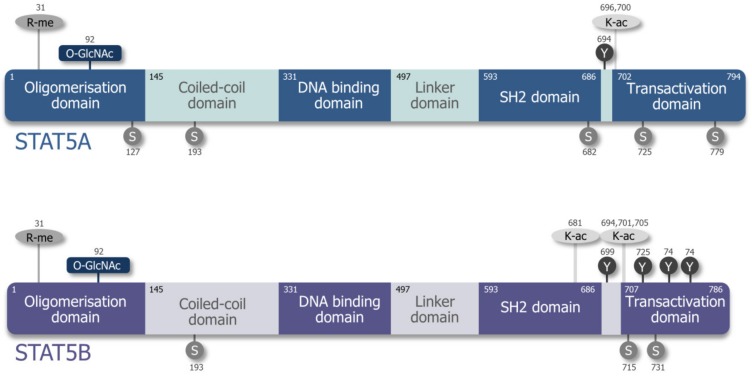
Differences in the domain structure and post-translational modifications of STAT5A and STAT5B. The protein structure of human STAT5A and STAT5B, including the most prominent Serine (S) and Tyrosine (Y) phosphorylation, Arginine methylation (R-me), and Lysine acetylation (K-ac), as well as O-GlcNAc sites, are shown.

**Figure 2 cancers-11-01726-f002:**
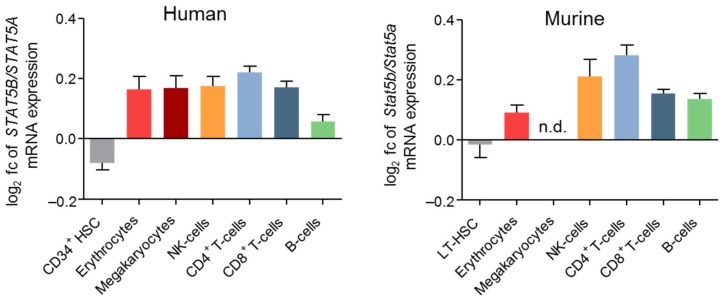
STAT5A and STAT5B mRNA expression levels in hematopoietic cells. Log2 fold change (fc) of STAT5B/STAT5A mRNA expression ratio of human (left) or murine (right) hematopoietic cells [77] using the human probes #212550_at, 212549_at, and 203010_at, and the murine probes #1421469_a_at and 1422103_a_at. (n.d., not determined).

**Figure 3 cancers-11-01726-f003:**
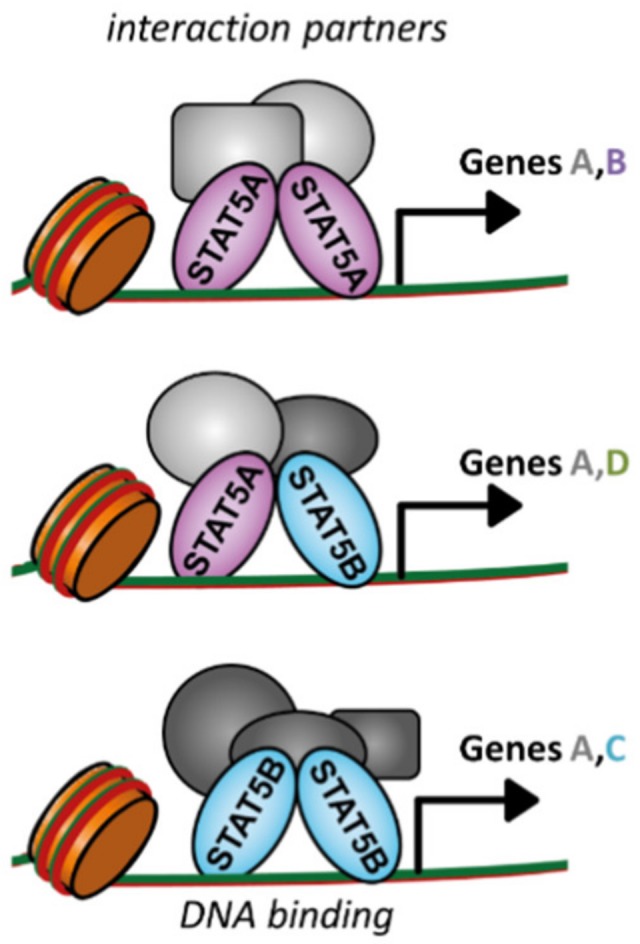
STAT5A and STAT5B hetero- and homodimers may induce different transcriptional programs. STAT5A and STAT5B may have unique DNA binding sites and induce different sets of target genes by interacting with distinct partners.

**Table 1 cancers-11-01726-t001:** *Stat5a/b*-deficient mouse models.

Deficiency		Phenotype	Reference
**STAT5A/B**	*Stat5a/b* ^ΔN^	truncated N-termini of *Stat5a* and *Stat5b* forms dimers but no tetramers smaller, infertile, less CD25^+^CD4^+^ T cells	Teglund et al. 1998 [56]
	*Stat5a/b^-/-^*	total body knockout perinatal lethal, 1–2% survivors dwarfism, anemia, reduced T- and NK cell numbers, block in pre–pro-B cell stage	Cui et al. 2004 [59]
	*Stat5a/b^DKI^*	N-termini mutations in *Stat5a* and *Stat5b* no tetramer formation less CD25^+^CD4^+^ and CD8^+^ T cells, less NK cells	Lin et al. 2012 [57]
	*Stat5a/b^fl/fl^*	loxP-sites spanning *Stat5a* and *Stat5b* for tissue-specific or inducible deletion—e.g., in the hematopoietic system:	Cui et al. 2004 [59]
		vav-Cre: Anemia, lymphopenia, reduced repopulation capacity (upon BM transplants)	Wang et al. 2015 [60]
		Tie2-Cre: Anemia, lymphopenia, reduced repopulation capacity (upon BM transplants)	Zhu et al. 2008 [61]
		Mx1-Cre: Reduced repopulation capacity (upon BM transplants)	Wang et al. 2009 [62]
**STAT5A**	*Stat5a* ^ΔN^	truncated N-termini of *Stat5a* forms dimers but no tetramers reduced prolactin signaling (mammary gland)	Teglund et al. 1998 [56]
	*Stat5a* ^−/−^	total body knockout of *Stat5a* failure in mammary gland formation and function	Liu et al. 1997 [9]
	*Stat5a^KI^*	N-termini mutation of *Stat5a* no STAT5A tetramers slight reduction of IL-2R expression on T cells	Lin et al. 2012 [57]
**STAT5B**	*Stat5b* ^ΔN^	truncated N-termini of *Stat5b* forms dimers but no tetramers dwarfism, reduced IGF-1 levels	Teglund et al. 1998 [56]
	*Stat5b* ^−/−^	total body knockout of *Stat5b* dwarfism, failure in GH signaling, reduced numbers of NK and T cells	Udy et al. 1997 [63]
	*Stat5b^KI^*	N-termini mutations of *Stat5b* no STAT5B tetramers same, but more pronounced as in *Stat5a^KI^*	Lin et al. 2012 [57]

**Table 2 cancers-11-01726-t002:** Hematopoietic STAT5A/B transgenic mouse models.

Transgene	Promoter	Phenotype	Reference
***Stat5b*-tg**	H-2K^b^ promoter and IgM enhancer (T, B, NK cells)	more CD8^+^ T cells ~12% thymic T cell lymphoblastic lymphoma (CD8^+^CD4^+^ or CD8^+^)	Kelly et al. 2003 [105,153]
	NOD background	75% CD8^+^ T cell lymphoblastic lymphoma	Chen et al. 2013 [154]
***Stat5b-CA*-tg**	Lck promoter and IgM enhancer	expansion of CD8^+^ and γδ T cells late emergence of clonal B cell lymphoma/leukemia (low incidence)	Burchill et al. 2003 [88]
	pre-BCR pathway defects	increased B-ALL incidence	Katerndahl et al. 2017 [89]
**cS5^F^**	Eµ enhancer (lymphoid specific)	increase of CD8^+^ T cells, late emergence of clonal B cell lymphoma/leukemia (low incidence)	Joliot et al. 2006 [129]
**MMTV-tTA TetO-cS5^F^**	cS5^F^-Tet-OFF under MMTV-LTR promoter	hematopoiesis unaffected cross to *Stat5a/b*^ΔN^ → expansion of granulocytes	Lin et al. 2013 [155]
***hSTAT5B***	*vav* promoter	mild expansion of CD8^+^ T cells no disease	Pham et al. 2018 [84]
***hSTAT5B*^N642H^**	*vav* promoter	aggressive CD8^+^ T cell neoplasia enhanced numbers of HSCs and lymphoid progenitors	Pham et al. 2018 [84]
**cS5^F lo^**	*vav* promoter	mild expansion of CD8^+^ T cells no disease	Maurer et al. 2019 [83]
**cS5^F hi^**	*vav* promoter	CD8^+^ T cell neoplasia correlating to PTCL-NOS enhanced numbers of HSCs and lymphoid progenitors	Maurer et al. 2019 [83]

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
