# Peer review of "STAT5A and STAT5B—Twins with Different Personalities in Hematopoiesis and Leukemia"

_cancers, 2019, doi:10.3390/cancers11111726_

Round 1

Reviewer 1 Report

In this review, a focus is made on STAT5 A and B and compared to twins with different personalities.

The title of the review nicely summarizes the different characteristics of these two proteins and thus the similar but also opposite roles highlighted.
This review describes the known roles of STAT5 A / B in a precise and well referenced way. It compares the importance of each STAT5 in normal hematopoiesis mainly in erythropoiesis, B differentiation and NK cell development but also in several hematological malignancies. The major role of STAT5 B is well described in the development of the different hematopoietic lineages and the participation of STAT5 A and B is discussed in the maintenance of the hematopoietic stem cell and its self-renewal. Their involvement in cancer processes focused here in the hematological malignancies is described through various leukemias thus presenting the pharmacological approaches of inhibition of STAT5 proteins that have been developed. The figures illustrate the words well and bring a lot of information and hypotheses.

Some informations should be clarified.

For example, in Figure 1 the legend explain arginine (R) methylation and Lysine (K) acetylation while in the picture there is R-me and K-ac.

Results in Figure 2 show in human and in mice the log2 of the ratio of STAT5B/STAT5A for different hematopoietic subpopulations. Authors wrote that STAT5B is expressed at higher levels compared to STAT5A (lines 156-157). Did this mean that the ratio is below 1.3 ?

The legend in Figure 2 should be consistent for the Y axis.

Line 428 : police size is different in the text

Author Response

Reviewer 1

In this review, a focus is made on STAT5 A and B and compared to twins with different personalities.

The title of the review nicely summarizes the different characteristics of these two proteins and thus the similar but also opposite roles highlighted.
This review describes the known roles of STAT5 A / B in a precise and well referenced way. It compares the importance of each STAT5 in normal hematopoiesis mainly in erythropoiesis, B differentiation and NK cell development but also in several hematological malignancies. The major role of STAT5 B is well described in the development of the different hematopoietic lineages and the participation of STAT5 A and B is discussed in the maintenance of the hematopoietic stem cell and its self-renewal. Their involvement in cancer processes focused here in the hematological malignancies is described through various leukemias thus presenting the pharmacological approaches of inhibition of STAT5 proteins that have been developed. The figures illustrate the words well and bring a lot of information and hypotheses.

Some informations should be clarified.

For example, in Figure 1 the legend explain arginine (R) methylation and Lysine (K) acetylation while in the picture there is R-me and K-ac.

We apologize for the misunderstanding and clarify. Descriptions in Figure legend 1 were changed into “Arginine methylation (R-me)”, “Lysine acetylation (K-ac)” and “O-linked N-acetylglucosamine (O-GlcNAc)”. In addition, we made minor changes to the locations of the depicted PTMs.

Results in Figure 2 show in human and in mice the log2 of the ratio of STAT5B/STAT5A for different hematopoietic subpopulations. Authors wrote that STAT5B is expressed at higher levels compared to STAT5A (lines 156-157). Did this mean that the ratio is below 1.3 ?

The reviewer pointed out correctly that STAT5B expression is increased by ~20-30% compared to STAT5A across several differentiated blood cell types. This data is supported by publications investigating the expression levels of STAT5A and STAT5B in specific cell types in more detail (e.g. Villarino et al. eLife 2016). We illustrated the log2 fold change ratios to highlight the opposing STAT5B/STAT5A ratio in HSCs compared to differentiated blood cells.

The legend in Figure 2 should be consistent for the Y axis.

We thank the reviewer for pointing out this mistake. We corrected the figure legend and updated the corresponding figure.

Line 428 : police size is different in the text

We thank the reviewer for pointing out this mistake and we corrected the font size.

Reviewer 2 Report

Maurer et al. propose a nice Review on STAT5 contribution to hematopoietic malignancies, with a special emphasis on the recently acknowledged major role of STAT5B over STAT5A.

This review is well documented and of high relevance for the STAT5 and hematological malignancy community. I would recommend minor revisions aiming to improve the quality of the text, which shows some weaknesses, as well as fill a few identified literature gaps, as follows:

Some proof-reading/editing is needed throughout to correct a number of typos and grammatical mistakes. Some are listed here: Line 12 and 37: replace “frequent” by “frequently” Line 41: should read “focuses” (not “focusses”) Line 51: “a” mammalian orthologue Throughout text and Table 1: replace “C’-terminus/termini” and “N’-terminus/termini” by “C-terminus/termini” and “N-terminus/termini” Lines 65-66: should read “The last 20 amino acids of STAT5A and the last 8 amino acids of STAT5B are unique to the respective proteins.” Line 77: should read “The critical tyrosine phosphorylation site for activation is Y694 in STAT5A and Y699 in STAT5B” Line 104: should read “A very recent study focusing on uSTAT5A and uSTAT5B in acute myeloid leukemia (AML) suggested that uSTAT5B is a key regulator of differentiation of AML cells” Line 105: should read “Isoform-specific interaction partners were identified in AML cell lines: uSTAT5A preferentially interacts with DBC1, while uSTAT5B preferentially interacts with ETV6” Line 123: period (“”.”) is missing after “stage” Line 146: replace “verify” by “confirmed” Line 152: “functions” (plural) Throughout: hyphens are improperly used in some places (B-cell on lines 193 and 199, NK-cells on lines 209, 210, 212, 213, 217, 220; T-cell on lines 223, 226, 228, 229, 230, 238; etc) Lines 217-219: sentence is unclear and should be rewritten Lines 219-220: should read “This may be explained by the higher expression level of STAT5B compared to that of STAT5A in NK cells” Lines 244-245: should read “These data generated form knockout mouse models do not completely reflect...” Line 263: should read “In line with these results, RNAi-mediated downregulation...” Line 294: should read “...increased in transgenic STAT5BN642H and STAT5B wild-type mice” Lines 307-308: should read “SOCS1 negatively regulates the levels of pYSTAT5A/B.” Lines 346-348: should read “Despite the phenotypic similarities of the cS5F vav mouse model compared to the STAT5BN642H transgenic mice, the latter disease model is far more drastic and aggressive. This pinpoints to a greater oncogenic potential of STAT5B compared to STAT5A.” Line 376: PIM1 and BCL2 (human genes) should be italicized Lines 400-401: should read “Forced expression of STAT5A led to downregulation of NPM-ALK through direct transcriptional inhibition” Lines 402-403: should read “Currently, NPM-ALK is the only oncogene for which an antagonistic function of STAT5A and STAT5B has been described.” Line 415: should read “The lead compound AC-3-19, which structure is based on salicylic acid,...” Lines 438-439: should read “Recently, ATP was found to bind to and inhibit the STAT5B SH2 domain [186], ....” Lines 454-455: should read “Whether the JAK/aurora kinase inhibition can effectively eradicate or ameliorate the STAT5BN642H-driven T-cell disease in vivo remains to be shown” Line 487: “vs.” should replaced by “VS” Line 846: reference 119 is missing Past and present tense should be used properly, e.g. present tense for state-of-the-art facts, and past tense when reporting experimental results. For instance, the paragraph between lines 115 to 120 uses the present tense while it should actually use the past tense (“STAT5A/BΔN proteins still formed dimers and bound DNA, but tetramer formation and complete target gene transcription were significantly impaired [21]. Hematopoiesis in Stat5aΔN, Stat5bΔN and Stat5a/bΔN mice was affected to a minor degree [54]. Likewise, tetramer formation was blocked in the Stat5a/bDKI (double knock-in) mouse model in which mutations were introduced into the N-termini of Stat5a or Both mouse models showed reduced numbers of natural killer (NK)-cells, while T-cell numbers were exclusively reduced in Stat5a/bΔN mice [55,56].”). Same between lines 131 and 136. Text should be revised throughout accordingly.

These two citations should be added in reference to “the non-redundant roles of STAT5A and STAT5B by affecting gene regulation” (line 69): Basham et al., 2008, Nucleic Acids Res. 36:3802-18; Kanai et al. 2014, PLoS One 9:e86790 (i.e. ref. [107]).

The description of combination therapies using epigenetic inhibitors targeting STAT5 cofactors (currently missing) is promising and highly relevant to oncogenic STAT5 and hematopoietic malignancies. A sentence should be added with the respective citations, as follows, at line 462 (after “...side effects cannot be anticipated.” and before “8. Conclusion”):

“Finally, combination therapies using epigenetic inhibitors targeting STAT5 cofactors, such as bromodomain and extra terminal inhibitors (BETi) and histone deacetylase inhibitors (HDACi), might offer alternative therapeutic approaches (Zhao et al. 2019 Neoplasia 21:82-92, Pinz et al. 2015 Nucleic Acids Res. 43:3524-45, Liu et al. 2014 Mol Cancer Ther. 13:1194-205, Fiskus et al. 2014 Mol Cancer Ther. 13:2315-27, Nguyen et al. 2011 Clin Cancer Res. 17:3219-32, Rascle et al. 2003 Mol Cell Biol. 23:4162-73).”

Author Response

Reviewer 2

Maurer et al. propose a nice Review on STAT5 contribution to hematopoietic malignancies, with a special emphasis on the recently acknowledged major role of STAT5B over STAT5A.

This review is well documented and of high relevance for the STAT5 and hematological malignancy community. I would recommend minor revisions aiming to improve the quality of the text, which shows some weaknesses, as well as fill a few identified literature gaps, as follows:

Some proof-reading/editing is needed throughout to correct a number of typos and grammatical mistakes. Some are listed here:

We thank the reviewer for pointing out these mistakes and we corrected them accordingly.

Line 12 and 37: replace “frequent” by “frequently” – thank you for pointing out this mistake.

Line 41: should read “focuses” (not “focusses”) – again, we are happy to be able to correct this mistake.

Line 51: “a” mammalian orthologue – we are sorry for this mistake and corrected it

Throughout text and Table 1: replace “C’-terminus/termini” and “N’-terminus/termini” by “C-terminus/termini” and “N-terminus/termini”. We corrected in line 61, 114, 119 and throughout Table 1.

Lines 65-66: should read “The last 20 amino acids of STAT5A and the last 8 amino acids of STAT5B are unique to the respective proteins.” We are happy to use this improved sentence.

Line 77: should read “The critical tyrosine phosphorylation site for activation is Y694 in STAT5A and Y699 in STAT5B” –we corrected the “are” to “is”

Line 104: should read “A very recent study focusing on uSTAT5A and uSTAT5B in acute myeloid leukemia (AML) suggested that uSTAT5B is a key regulator of differentiation of AML cells” – we are happy to use the corrected syntax

Line 105: should read “Isoform-specific interaction partners were identified in AML cell lines: uSTAT5A preferentially interacts with DBC1, while uSTAT5B preferentially interacts with ETV6” – thank you, we corrected accordingly but skipped the word “preferentially”, as it changes the meaning of the sentence

Line 123: period (“”.”) is missing after “stage” – we are sorry for this mistake and inserted the punctuation mark.

Line 146: replace “verify” by “confirmed” – thank you for this suggestion, we changed the word accordingly.

Line 152: “functions” (plural) - corrected

Throughout: hyphens are improperly used in some places (B-cell on lines 193 and 199, NK-cells on lines 209, 210, 212, 213, 217, 220; T-cell on lines 223, 226, 228, 229, 230, 238; etc) – we changed all NK-cells, T-cells and B-cells to NK cells, T cells and B cells

Lines 217-219: sentence is unclear and should be rewritten.

The rewritten sentence reads as follows: STAT5A/B dimers were sufficient for NK cell development, whereas tetramers were needed for maturation [94].

Lines 219-220: should read “This may be explained by the higher expression level of STAT5B compared to that of STAT5A in NK cells” – thank you for this correction which we directly applied on the respective sentence.

Lines 244-245: should read “These data generated form knockout mouse models do not completely reflect...” – the reviewer was totally right that the word “murine” has to be skipped

Line 263: should read “In line with these results, RNAi-mediated downregulation...” - corrected

Line 294: should read “...increased in transgenic STAT5BN642H and STAT5B wild-type mice” – as suggested, we added “wild-type” (wt) to clarify the meaning of the sentence.

Lines 307-308: should read “SOCS1 negatively regulates the levels of pYSTAT5A/B.” We are happy to use this improved sentence in the revised version of the manuscript.

Lines 346-348: should read “Despite the phenotypic similarities of the cS5F vav mouse model compared to the STAT5BN642H transgenic mice, the latter disease model is far more drastic and aggressive. This pinpoints to a greater oncogenic potential of STAT5B compared to STAT5A.” – we incorporated the correction of the reviewer

Line 376: PIM1 and BCL2 (human genes) should be italicized - corrected

Lines 400-401: should read “Forced expression of STAT5A led to downregulation of NPM-ALK through direct transcriptional inhibition” – as suggested, we exchanged “by its” to “through”

Lines 402-403: should read “Currently, NPM-ALK is the only oncogene for which an antagonistic function of STAT5A and STAT5B has been described.” – as suggested, we exchanged “where” to “for which”

Line 415: should read “The lead compound AC-3-19, which structure is based on salicylic acid,...” – thank you for improving this sentence.

Lines 438-439: should read “Recently, ATP was found to bind to and inhibit the STAT5B SH2 domain [186], ....” – thank you, we corrected the sentence accordingly.

Lines 454-455: should read “Whether the JAK/aurora kinase inhibition can effectively eradicate or ameliorate the STAT5BN642H-driven T-cell disease in vivo remains to be shown” – thank you, we corrected the sentence accordingly.

Line 487: “vs.” should replaced by “VS” – we apologize for this mistake and corrected the initials to VS.

Line 846: reference 119 is missing – we inserted the correct reference details.

Past and present tense should be used properly, e.g. present tense for state-of-the-art facts, and past tense when reporting experimental results. For instance, the paragraph between lines 115 to 120 uses the present tense while it should actually use the past tense (“STAT5A/BΔN proteins still formed dimers and bound DNA, but tetramer formation and complete target gene transcription were significantly impaired [21]. Hematopoiesis in Stat5aΔNStat5bΔand Stat5a/bΔmice was affected to a minor degree [54]. Likewise, tetramer formation was blocked in the Stat5a/bDKI (double knock-in) mouse model in which mutations were introduced into the N-termini of Stat5a or Both mouse models showed reduced numbers of natural killer (NK)-cells, while T-cell numbers were exclusively reduced in Stat5a/bΔmice [55,56].”). corrected

Same between lines 131 and 136. corrected

Text should be revised throughout accordingly.

We corrected from present to past tense throughout the text highlighted by track-changes.

These two citations should be added in reference to “the non-redundant roles of STAT5A and STAT5B by affecting gene regulation” (line 69): Basham et al., 2008, Nucleic Acids Res. 36:3802-18; Kanai et al. 2014, PLoS One 9:e86790 (i.e. ref. [107]).

We added the references manually with #107 and #195.

The description of combination therapies using epigenetic inhibitors targeting STAT5 cofactors (currently missing) is promising and highly relevant to oncogenic STAT5 and hematopoietic malignancies. A sentence should be added with the respective citations, as follows, at line 462 (after “...side effects cannot be anticipated.” and before “8. Conclusion”):

“Finally, combination therapies using epigenetic inhibitors targeting STAT5 cofactors, such as bromodomain and extra terminal inhibitors (BETi) and histone deacetylase inhibitors (HDACi), might offer alternative therapeutic approaches (Zhao et al. 2019 Neoplasia 21:82-92, Pinz et al. 2015 Nucleic Acids Res. 43:3524-45, Liu et al. 2014 Mol Cancer Ther. 13:1194-205, Fiskus et al. 2014 Mol Cancer Ther. 13:2315-27, Nguyen et al. 2011 Clin Cancer Res. 17:3219-32, Rascle et al. 2003 Mol Cell Biol. 23:4162-73).”

We thank the reviewer for the very thoughtful comment and inserted the additional sentence. References were inserted at the end of the reference list with #196-201.